# OpenReview forum: "AIPO : Adaptive Intent-driven Preference Optimization"
_ICLR.cc/2026/Conference — ICLR 2026 Conference Desk Rejected Submission_

### Official Review · Reviewer_pf2h · 2025-10-22

**Soundness:** 3
**Presentation:** 2
**Contribution:** 2
**Rating:** 4
**Confidence:** 4

**Summary:**

The paper proposes **A-IPO (Adaptive Intent-Driven Preference Optimization)**, an extension of DPO that conditions the reward on a latent intent variable $I$ inferred from the prompt. The method uses:
1. prompt decomposition via LLM calls,
2. retrieval from Wikipedia through Pinecone,
3. fact-checking via Anah-v2, and
4. an intent classifier $q_\phi(I|x)$ guiding an augmented DPO loss with a similarity term $\lambda\mathrm{sim}(y,I)$.

Theoretical results (Lemma 5.2, Theorem 5.3) show that this intent-conditioned term increases preference margins and decreases pairwise NLL. Experiments on GPT-2 Large and Pythia-2.8B show consistent gains on author-curated datasets (REAL-PREF, ATTACK-PREF, GlobalOpinionQA-Ext).

While the direction is interesting, the pipeline adds substantial computational overhead (multiple LLM calls, retrieval/fact-checking, and an auxiliary classifier) without cost reporting. It also doesn’t compare to simpler inference-time intent handling.

**Strengths:**

- Addresses the limitation of DPO’s single-preference assumption and aims for pluralistic alignment.
- Provides a clean modular pipeline (intent inference + DPO with a similarity term).
- Offers formal though limited proofs of margin and NLL improvement.
- Shows consistent metric gains (Table 1, 2) and reasonable ablations.
- Includes detailed appendices with templates and dataset stats.

**Weaknesses:**

1. **Mathematical correctness and internal consistency**
   - **Similarity term outside the logit.** In the BT/PL setup, probabilities should be $\sigma(\text{logit})$. Eqs. (38), (39) for example, add $\lambda[\mathrm{sim}(y_w,I)-\mathrm{sim}(y_l,I)]$ **after** the sigmoid/logit step, which can push the expression outside $[0,1]$ and isn’t a valid probability transformation. In Appendix D and Lemma 5.2, the same term is treated as a **logit shift** (i.e., inside the log-odds). This mismatch makes the central training objective mathematically incoherent, and I am not sure if the probability model is valid.
   - **Theorem 5.1 statement.** The LHS is written as $r(x,y)$ (no $I$) while the RHS depends on $I$; the proof immediately “fixes $(x,I)$,” suggesting the intended statement is $r(x,y,I)$ (or “for each fixed $I$”). This should be fixed to avoid confusion about what is being identified.
   - **Notation drift.** $D_{KL}$ vs.\ $KL$; $q_\phi(I|x)$ in some places vs.$q_\tau(I|x)$ elsewhere; $\mathcal D$ called a “dataset” but used as a sampling distribution $x\sim \mathcal D$. These are small individually, but in aggregate they are a bad look for the paper.

2. **Undefined core component drives the main effect**
   - The paper never specifies how $\mathrm{sim}(y,I)$ is computed: What text encoders are used for $y$ and for $I$? Is $I$ represented by a discrete embedding or a continuous vector? What is the **range/scale** of $\mathrm{sim}$ (bounded in $[0,1]$ or unbounded cosine, etc.)? Is there normalization or a temperature? Do gradients back-prop through the response encoder? Because $\lambda$ scales $\mathrm{sim}$ directly, the **numerical range** of $\mathrm{sim}$ critically determines the margin effect and stability. Without these details, the method is not reproducible and the ablations are hard to interpret.

3. **Ambiguity about intent supervision**
   - The main text defines an element-wise BCE over $K$ intentions (multi-label), but the experimental setup later describes a single-label softmax classifier for the intention module. These correspond to **different problem formulations** (multi-label vs single-label). The paper should commit to one, explain why, and reflect that choice consistently in the loss and implementation.

4. **Dataset–objective coupling that favors the proposed method**
   - The REAL-PREF curation template instructs the generator to (i) craft a prompt that **explicitly embeds an intention**, (ii) produce an “accept” response that **honors** that intention, and (iii) a “reject” response that **contradicts** it. ATTACK-PREF similarly constructs synthetic corruptions. Then A-IPO optimizes a loss that **directly rewards intent–response similarity**. This structural alignment between how negatives are generated and how the model is trained very likely advantages A-IPO over baselines that do not explicitly encode an intent similarity term. This doesn’t invalidate the gains, but limits what we can conclude about generalization to less templated, naturally occurring preferences.

5. **Baseline tuning and scope**
   - All baselines reportedly share the same hyperparameters. Prior work shows DPO/GDPO often require **method-specific** $\beta$ and optimizer schedules; a one-size setting can understate their performance. In addition, experiments are limited to GPT-2 Large and Pythia-2.8B; showing results on at least one larger open model would improve external validity.

6. **Compute and systems cost not quantified; missing inference-time alternative**
   - The method adds multiple moving parts around DPO: several LLM calls for prompt decomposition and intent extraction, retrieval (Pinecone) and sentence-level fact-checking (Anah-v2), plus an intent classifier and the augmented training loop. Even if some steps are precomputed offline, this **increases engineering complexity and retraining cost** whenever domains shift. The paper acknowledges “computational overhead” but reports no wall-clock, GPU-hour, or memory numbers.
   - For the cultural/knowledge intents considered, a strong **inference-time baseline** (detect intent at serve time, retrieve evidence, and re-rank or constrain decoding using a similarity score) could deliver similar gains **without retraining**. The absence of this comparison leaves the practical cost–benefit unanswered.

8. **Clarity gaps that impede reproduction**
   - The prior $p(I)$ is not specified (uniform? learned?), and there is no check for posterior collapse or sensitivity to the prior.
   - There are minor grammar/formatting issues (e.g., subject–verb agreement; missing spaces around symbols), which cumulatively reduce polish and increase cognitive load for the reader.

**Questions:**

Every item raised in the *Weaknesses* section can be viewed as a question for the authors.
I may well be mistaken on several of these points, and I would sincerely appreciate clarification or correction wherever appropriate.
If the authors can address or resolve even part of these concerns—whether by showing that I misunderstood something or by providing additional detail—it would be very helpful.

That said, my top three concerns I’d especially like to hear more about are:

1. **The probabilistic modeling issue:** whether the similarity term’s placement outside the logit in Eqs. (8)/(39)/(40) is indeed an error or whether there’s a principled justification that I missed.
2. **The dataset generation bias:** how the authors think about potential coupling between their curated templates and the method’s design, and whether they have evidence of generalization beyond that distribution.
3. **The scale limitation:** whether they have tried or plan to test A-IPO on larger models (at least 7B parameters) to verify scalability and the claimed improvements on pluralistic alignment.

I offer these comments in a constructive spirit and look forward to the authors’ perspective on these points and on any others mentioned in the *Weaknesses* section.

**Details Of Ethics Concerns:**

* Cultural/identity framing. The REAL‑PREF curation explicitly encodes cultural/religious intents (e.g., names, taboos) and generates “accept/reject” labels accordingly (Appendix I.5). Even with guardrails (“Do not assert religion based solely on name…”, I.4), risks of stereotyping or misattribution remain, especially if deployed widely.

* LLM‑generated, LLM‑judged data. Reliance on GPT‑4 both to generate and judge may import opaque biases; the paper does not report annotator demographics/compensation or human validation rates and agreements. Clearer data statements and human‑in‑the‑loop evaluations are recommended.

---

> ### Author Response · Authors · 2025-11-20
> **Response - W1**
>
> ## W1--Mathematical correctness and internal consistency
>
> ### W1-1. Similarity term outside the logit
>
> Thank you for pointing this out issue. The probability model we *intend* to use, and the one that is actually used in the theoretical analysis (Corollary D.3, Lemma 5.2 and Theorem 5.3) is
>
> $\operatorname{logit} \Pr(y_w \succ y_l \mid x, I)
> = \Delta' := \beta \Delta \log\text{ratio} + \lambda \Delta\text{sim}$,
>
> where
>
> $\Delta \log\text{ratio}
> = \log \frac{\pi_\theta(y_w\mid x,I)/\pi_{\text{ref}}(y_w\mid x,I)}{\pi_\theta(y_l\mid x,I)/\pi_{\text{ref}}(y_l\mid x,I)},
> \quad
> \Delta\text{sim}
> = \text{sim}(y_w,I) - \text{sim}(y_l,I)$
>
> This is exactly the form stated after Theorem 5.1, and it is the setting on which Lemma 5.2 and Theorem 5.3 are based.
>
> Conceptually, the similarity term should *always* appear as a feature **inside** the BT logit, rather than being added outside the sigmoid as an extra probability-level term. In the current main text, Eqs.~(8)/(39)/(40) mistakenly place $\lambda\Delta\text{sim}$ outside $\log\sigma(\cdot)$ due to a typing error, breaking this correspondence. We appreciate your careful reading in this regard.
>
> In the revision, we will make the following changes to remove the ambiguity and align with the implementation:
>
> 1. **Correct the equations.** We will rewrite Eqs. (8), (39), and (40) in the unified form
>
>    $\Pr(y_w \succ y_l \mid x,I) = \sigma \bigl(\beta \Delta \log\text{ratio} + \lambda \Delta\text{sim}\bigr)$, which is fully consistent with the ELBO derivation in Eq. (3) and the derivations in Appendix~C.
>
> 2. **Make the implementation form explicit in the theory section.** Right after Lemma 5.2, we will explicitly state that the implementation uses this “logit-shift’’ form, i.e., $\lambda\Delta\text{sim}$ is directly added to the DPO logit *before* applying the sigmoid, so that the probability model exactly matches the theoretical analysis.
>
> We confirm that this is indeed an issue at the level of notation/typesetting, rather than a fundamental inconsistency in the underlying BT/PL likelihood. The core probability model is coherent and is implemented in the logit-shift form described above.
>
>
> ---
>
> ## W1-2. Theorem 5.1 statement
>
> Thank you for carefully pointing it out. What we intend to state is the following:
>
> $r(x,y,I) = \beta \log \frac{\pi(y \mid x,I)}{\pi_{\text{ref}}(y \mid x,I)} + \lambda\,\text{sim}(y,I) + b(x,I)$
>
> which should hold for every fixed pair $(x,I)$.
>
> In the camera-ready version, we will:
>
> - explicitly condition on $I$ on both sides of the equality in Theorem 5.1, writing $r(x,y,I)$ rather than $r(x,y)$; and
> - clarify in the surrounding text that the theorem is applied pointwise for each fixed $(x,I)$, which is exactly consistent with the proof’s “fix $(x,I)$ and define the tilted policy $\pi(\cdot\mid x,I)$’’ step.
>
> These changes are purely presentational and do not affect any subsequent results.
>
> ---
>
> ## W1-3. Notation drift
>
> Thank you for carefully flagging these notation issues. In the revisied version, we will:
>
> - consistently use $\mathrm{KL}(\cdot\Vert\cdot)$ to denote the KL divergence throughout the paper;
> - consistently use $q_\tau(I\mid x)$ as the notation for the variational posterior, and remove occurrences of $q_\tau(I\mid z)$; and
> - reserve $\mathcal{D}$ strictly for the empirical dataset, and introduce a separate symbol (e.g., $\mathcal{P}_X$) for the underlying prompt distribution.
>
> These are “surface-level’’ cleanups, but they will help avoid unnecessary distraction from the core ideas.

---

> ### Author Response · Authors · 2025-11-20
> **Response - W2**
>
> ## W2--Undefined core component drives the main effect
>
>
> In our implementation, $\mathrm{sim}(y, I)$ is computed from:
> 1. the latent intent embedding produced by the intention module (Sec.4.2), and
> 2. a sequence-level embedding of the response $y$ obtained from the final hidden layer of the network.
>
> Formally, the predicted intent ($I$) indexes into a trainable embedding table $E$, giving
> $z_I = E[I]$. The response ($y$) is passed through the shared encoder, and its final-layer token representations are mean-pooled to obtain $h_y$.
>
>
> After $\ell_2$-normalization, we define the similarity function: $\mathrm{sim}(y, I) = \frac{1 + \langle \hat{z}_I, \hat{h}_y \rangle}{2} \in [0, 1]$, so that: $\Delta\mathrm{sim} = \mathrm{sim}(y_w, I) - \mathrm{sim}(y_l, I) \in [-1, 1]$,
> and the term $\lambda \Delta\mathrm{sim}$ acts as a bounded logit shift inside the Bradley–Terry likelihood used by A-IPO.
> Gradients from this term back-propagate through both $z_I$ and $h_y$, so the shared encoder and the intent embedding table
> are trained jointly with the preference objective. We will clarify this detail in the revised manuscript.

---

> ### Author Response · Authors · 2025-11-20
> **Response - W3**
>
> ## W3--Ambiguity about intent supervision
>
> Thanks for pointing this out.
>
> - At the theoretical level, we wanted to allow the possibility that a single sample might activate multiple intent categories, which is why Eq. (5) is written as a multi-label BCE over $K$ binary indicators.
> - In our concrete implementation for this paper, however, each sample is annotated with exactly one *core intent*, i.e., a single dominant intent label. Accordingly, we actually use a single-label softmax classifier with standard cross-entropy over $K$ classes, where each prompt has one ground-truth label $I^\star$.
>
> To avoid confusing readers, in the revision we will:
>
> - explicitly state in Sec. 4.2 and Appendix G.1.4 that the single-label softmax classifier is the default implementation used in this paper;
> - move the multi-label BCE formulation to the appendix as a natural extension for potential future datasets with multi-intent annotations; and
> - ensure that all notation related to $p_\phi(I\mid x_{\text{con}})$ and its loss consistently uses the single-label form for the experiments we report.
>
> This will align the supervision scheme and the implementation at the formal level.

---

> ### Author Response · Authors · 2025-11-20
> **Response - W4**
>
> ## W4--Dataset–objective coupling that favors the proposed method
>
> We thank the reviewer for raising the important issue of dataset–objective coupling. To clarify, the core motivation behind A-IPO is to develop a preference optimization framework that can systematically extract and leverage implicit intent signals present in prompts and data, including cultural and/or other context-dependent intents, during preference learning. This explicit modeling of intent differentiates A-IPO from previous preference optimization methods, which rarely account for intent and therefore struggle to effectively address complex, diverse, and culturally nuanced alignment scenarios. In this light, A-IPO is designed to overcome limitations of existing approaches, whose applicability is often restricted in real-world, heterogeneous settings precisely because they do not utilize or surface latent intent. By directly targeting intent extraction and integration, A-IPO is inherently better suited for scenarios that demand sensitivity to the kinds of intent variation found in open, multi-cultural environments.
>
> Given these objectives, it is essential to employ benchmarks that meaningfully test a method’s ability to discern and incorporate implicit intent when optimizing preferences. However, to the best of our knowledge, there are no existing preference optimization benchmarks that robustly operationalize these criteria or systematically evaluate intent sensitivity. This lack of appropriate evaluation standards motivated us to design *REAL-PREF* and *ATTACK-PREF* as new benchmarks specifically constructed to provide challenging, intent-sensitive evaluation environments. In summary, the introduction of the *REAL-PREF* and *ATTACK-PREF* benchmarks is not to unfairly advantage A-IPO, but to fill a gap in existing evaluation and to enable rigorous testing of methods that address implicit intent, which we argue is crucial for robust, real-world preference optimization.
>
> To further address concerns about potential bias from relying solely on our own constructed datasets, we also include experiments on *GlobalOpinionQA-Ext*, which is derived by extending an independent, third-party benchmark rather than curated entirely by us. Notably, on this externally sourced dataset, A-IPO continues to consistently outperform DPO and GDPO across Win-Rate, ICS, RIC, and RS metrics (see Table~1). This demonstrates that the advantages of A-IPO are not confined to our own benchmarks, but extend to settings beyond our own dataset construction.
>
> To clarify, the empirical evaluation of A-IPO aims at tasks where preference pairs distinctly separate intent-aligned from intent-violating responses. This focus aligns with labeling principles in many practical RLHF datasets. Nonetheless, we would like to emphasize that the *REAL-PREF* and *ATTACK-PREF* benchmarks are a natural choice for methods designed to model intent explicitly. We will make this explicit in our revision and add to the **Limitations** section a discussion noting that these benchmarks are more favorable to intent-aware and/or culture-specific settings.

---

> ### Author Response · Authors · 2025-11-20
> **Response - W5**
>
> ## W5--Baseline tuning and scope of LLMs
>
> ### Baseline Tuning
> **All experimental settings of A-IPO closely follow those established in the DPO and GDPO papers**, as well as their publicly available source codes, to ensure a fair and consistent comparison across methods. **Specifically, existing implementations as well as the published work of DPO and GDPO baselines are evaluated using the same experimental setups, including $\beta = 0.1$ and identical optimizer schedules.** While we acknowledge that exhaustive hyperparameter tuning for baseline methods could potentially improve their absolute performance, such efforts entail significant computational overhead. To ensure fairness and reproducibility, we therefore adhere to the experimental configurations recommended by the authors of DPO and GDPO.
>
> Experimental evaluation shows that A-IPO achieves substantial improvements on multiple key metrics (for example, on *REAL-PREF*, the Win-Rate improvement over GDPO can reach +9.5, and ICS can improve by +12.6), which makes it relatively unlikely that hyperparameter tuning alone would eliminate the gaps entirely.
>
> ### Scope of LLMs
> Similar to the baseline models, for comparative evaluation, we used widely used medium-scale open-source models (GPT-2 Large and Pythia-2.8B) for evaluating A-IPO. However, the algorithmic form of A-IPO is architecture-agnostic and can, in principle, be applied directly to 7B+ models. In the revised manuscript, we will add a theoretical justification for the scalability of A-IPO to larger 3B+ and even latest MoE based LLMs.

---

> ### Author Response · Authors · 2025-11-21
> **Resonse - W6**
>
> ## W6--Compute and systems cost; inference-time baseline
>
> ### W6-1. Compute and systems cost
>
> For the REAL-PREF benchmark, which consists of 7,303 preference pairs, model training with A-IPO using GPT-2-Large (774M parameters), a global batch size of 8–16, and a maximum sequence length of 1,024 on 2×A100 GPUs, requires approximately 10–12 minutes of wall-clock time per epoch. Empirically, we find that A-IPO introduces only a modest increase in training duration relative to DPO, corresponding to an overhead of approximately 10–15\%. This additional cost primarily arises from a single intent embedding lookup and an extra dot product operations for computing $\text{sim}(y, I)$.
>
> The computational requirements of the BERT-base intention classifier are substantially lower. For this component, a single training epoch (batch size 16, maximum sequence length 512) is completed in approximately 2 minutes on the same hardware, making its cost negligible compared to that of language model fine-tuning.
>
> All other preprocessing operations, including prompt decomposition, retrieval, and sentence-level fact-checking, are conducted offline, cached, and therefore do not contribute to the per-epoch training time. More concretely, large language model (LLM)-based prompt decomposition, retrieval through Pinecone, and sentence-level fact-checking with Anah-v2 are primarily employed during dataset construction and intent annotation phases (see Sec 4.2 and Appendix I); during preference training itself, only precomputed $x_{\text{con}}$ and intent labels are used.
>
>
> ### W6-2. Inference-time baseline at serving time
>
> We appreciate the reviewer’s suggestion of an inference-time baseline that detects intent at serving time. However, we argue that purely inference-time approach is fundamentally limited in several ways:
>
> 1. **No change to the underlying policy distribution.**
>    Inference-time intent detection and re-ranking operate *post hoc* on samples drawn from the base policy $\pi_\theta$. As a result, if $\pi_\theta$ assigns very low probability mass to intent-consistent outputs (e.g., for minority or underrepresented intents), the re-ranking can only select among a small set of **suboptimal candidates** already proposed by the model.
>
> 2. **Inability to correct systematic blind spots.**
>    Suppose the base model almost never generates acceptable answers for certain intents (e.g., underrepresented regions/cultures/minorities or niche expertise). In that case even an optimal re-ranking strategy cannot select an answer that the model never produces intent-aware inference-time heuristics merely choose “the best among several bad options.”
>
> 3. **Lack of gradient-driven adaptation from preference data.**
>    Inference-time methods typically treat intent similarity as a *scoring* or *filtering* functionas they do not back-propagate this signal into the model parameters.
>
> 4. **Serving-time efficiency and robustness.**
>    Inference-time intent detection with retrieval and re-ranking must be executed for **every request** at deployment, which increases latency and serving cost (multiple forward passes, retrieval calls, and scoring).
>
> Accordingly, we will consider these revisions in the revised manuscript.

---

> ### Author Response · Authors · 2025-11-21
> **Response - W7**
>
> ## W7--Clarity gaps and prior specification
>
> We acknowledge the omission and will clarify in the manuscript that a uniform discrete prior is used over the $K$ intent categories, i.e., $p(I=k) = 1/K$. This choice imposes no additional preference among intents and simplifies the KL term in Eq. (3). We did not observe posterior collapse with this setting.
>
> We also appreciate the feedback regarding presentation. For the final version, we will systematically address grammatical consistency, mathematical formatting, and clarity of figures and captions.

---

> ### Author Response · Authors · 2025-11-21
> **Response - Q1, Q2, and Q 3**
>
> ### For Q1, please refer to Response - W1
>
> ### For Q2, please refer to Response - W4
>
> ### For Q3, please refer to Response - W5
>
> We hope these clarifications will assist the reviewer in reconsidering the evaluation of our work.

---

> ### Author Response · Authors · 2025-11-27
> **Rebuttal Feedback**
>
> **Dear Reviewer pf2h**,
>
> We thank you for your thoughtful and constructive feedback. We have carefully addressed all questions and points raised, providing clarifications, additional analyses, and revisions where appropriate, and we hope our responses satisfactorily resolve the concerns. Should there be any further questions or points requiring elaboration, please let us know — we would be happy to provide additional clarification. In light of these clarifications and improvements, we hereby and respectfully request a re-evaluation of the paper’s scores.
>
> Thanks

---

### Official Review · Reviewer_kpR4 · 2025-10-30

**Soundness:** 3
**Presentation:** 3
**Contribution:** 3
**Rating:** 8
**Confidence:** 2

**Summary:**

The paper introduces Adaptive Intent-driven Preference Optimization (A-IPO), a framework for aligning large language models that tackles   several shortcomings of Direct Preference Optimization (DPO). The authors contend   that DPO’s use of a single, averaged preference signal tends to amplify majority opinions, overlook subtle user intentions, and weaken the model’s resilience to adversarial inputs.

**Strengths:**

The paper’s greatest strength lies in its clarity and focus. It pinpoints a specific, well-recognized problem—DPO’s “tyranny of the majority”—and presents a solution that is both intuitive and well-motivated. The introduction clearly articulate the motivation and high-level idea, setting an excellent standard for exposition.

The empirical analysis is thorough and convincing. The authors introduce targeted benchmarks—REAL-PREF and ATTACK-PREF—to properly evaluate the model’s claims. Results demonstrate clear improvements over strong baselines, while the ablation study highlights the necessity of each component within A-IPO.

**Weaknesses:**

The framework’s effectiveness largely depends on the intention module, which is trained in a supervised fashion using a cross-entropy loss on ground-truth binary intent labels. This design implies that A-IPO requires access to a dataset annotated with high-quality, discrete intent labels. While the paper criticizes GDPO for its reliance on belief or group partitioning, it remains unclear how A-IPO’s dependence on explicit intent supervision is more scalable or less costly.

**Questions:**

Maybe I have missed this, but how is the sim(y, I) function (Eq 7) implemented?

---

> ### Author Response · Authors · 2025-11-20
> **Response---W1**
>
> ## W1: Annotation cost and scalability of intent supervision
>
> The key point is that A-IPO does **not** require an external or fundamentally new form of annotation; instead, it is designed to **surface intent that is already implicit in the data**, leveraging cues or annotation layers often embedded (explicitly or implicitly) in existing alignment/safety datasets. We will explain this in more detail below.
>
> ### 1. **Comparison to GDPO's supervision requirements:**
> The primary distinction between A-IPO and GDPO lies in the nature and scalability of annotation required for each framework. GDPO fundamentally depends on the provision of *external belief information*, necessitating the construction of dataset-specific group partitions based on prior knowledge or custom definitions of "belief" or "group". These belief groupings must be supplied independently of the dataset, often through synthetic rules, statistical stratification, or manual curation. Consequently, the annotation and setup overhead for GDPO scales with the number of domains or belief axes, resulting in substantial manual effort and limited flexibility.
> Moreover, such externally imposed belief partitions may not align with the inherent intent or cultural nuances of the data and/or application scenario.
> A-IPO obviates the need for fragile, externally defined groupings by leveraging intent signals latent or explicit in the data, and is thus formally more suitable and scalable for open-domain, culturally nuanced, or adversarial settings compared to group- or belief-dependent approaches like GDPO. We claim, A-IPO is **geared towards extracting intent that is already latent in the data and/or input prompt**, aiming to unify and systematize this process. Rather than additional beliefs, it uses a **shared intent taxonomy** (e.g., 6 domains and 231 intent categories for *REAL-PREF*) that can be applied for entire *REAL-PREF* data. Concretely, each example is linked to its underlying intent commonly present or inferable within data, such as topic, cultural intricacies, task type, and even other high-level categories. Thus, A-IPO’s annotation effort is not only lower compared to the group construction needed by GDPO, but is also **inherently more scalable** because intent can be surfaced and reused across datasets.
>
>
> ### 2. **Scalable annotation via semi-automatic intent inference:**
> A-IPO's annotation process is explicitly designed for scalability by reducing dependency on exhaustive manual labeling and leveraging automation wherever possible. Our pipeline operates as follows:
> - A state-of-the-art (SOA) proprietary LLM generates candidate intent labels at scale, exploiting correlations and features already inherent in large datasets.
> - Human annotators are only required to review and correct a small, representative subset to verify label quality. Importantly, this manual step remains constant and does not increase with the overall dataset size.
> - A lightweight intent classifier is trained on this curated subset and then deployed to automatically pseudo-label the vast remainder of the dataset, allowing intent annotation to seamlessly scale as new data is added.
>
> This approach sharply contrasts with frameworks like GDPO, where group or belief structures require bespoke, manual construction for each dataset or domain. In A-IPO, once the intent taxonomy and classifier are established, annotating additional data incurs only minimal incremental effort, enabling efficient expansion to massive or continually evolving datasets with little manual overhead.
>
>
> ### 3. **Robustness and practicality:**
> Crucially, A-IPO is *robust to noisy or imperfect intent labels* because it incorporates inferred intent directly into the model learning process, using it as a reward adjustment to improve the margin between preferred and dispreferred responses. This design enables the model to learn more discriminative preference boundaries even when the intent signal is ambiguous or noisy, enhancing its robustness in practical settings compared to GDPO, which is more reliant on externally defined groupings. Empirically (Sec. 7.3), we observe that A-IPO continues to deliver strong gains under challenging or imperfect annotations, highlighting the effectiveness of extracting and integrating implicit intent from data during learning.
>
>
> In summary, whereas GDPO’s core reliance is on externally provided belief inputs often requiring extensive, dataset-specific manual construction, A-IPO is explicitly designed to *dig out* the intent already present or latent in real-world data. Our approach is **shared, modular, and linearly scalable** with dataset size, making it clearly more suitable for generalization across open-domain, cultural, or adversarial settings than group-/belief-specific frameworks such as GDPO.

---

> ### Author Response · Authors · 2025-11-20
> **Response Q1**
>
> ## Q1--Implementation details of $\mathrm{sim}(y, I)$
>
>
> In our implementation, $\mathrm{sim}(y, I)$ is computed from:
> 1. the latent intent embedding produced by the intention module (Sec.4.2), and
> 2. a sequence-level embedding of the response $y$ obtained from the final hidden layer of network.
>
> Formally, the predicted intent $(I)$ indexes into a trainable embedding table $E$, giving
> $z_I = E[I]$. The response $(y)$ is passed through the shared encoder, and its final-layer token representations are mean-pooled to obtain $h_y$.
>
>
> After $\ell_2$-normalization, we define the similarity function: $\mathrm{sim}(y, I) = \frac{1 + \langle \hat{z}_I, \hat{h}_y \rangle}{2} \in [0, 1]$, so that: $\Delta\mathrm{sim} = \mathrm{sim}(y_w, I) - \mathrm{sim}(y_l, I) \in [-1, 1]$,
> and the term $\lambda \Delta\mathrm{sim}$ acts as a bounded logit shift inside the Bradley–Terry likelihood used by A-IPO.
> Gradients from this term back-propagate through both $z_I$ and $h_y$, so the shared encoder and the intent embedding table
> are trained jointly with the preference objective. We will clarify this detail in the revised manuscript.

---

### Official Review · Reviewer_VQt8 · 2025-11-01

**Soundness:** 3
**Presentation:** 4
**Contribution:** 3
**Rating:** 6
**Confidence:** 4

**Summary:**

This work proposes A-IPO, a framework that incorporates user-specific latents in reward modeling via an intention module, and demonstrate both empirically and theoretically that this increases separation between preferred and dispreferred responses for diverse users. This work introduces two new datasets, REAL-PREF and ATTACK-PREF, and extends the existing GlobalOpinionQA dataset, to benchmark real-world and adversarial preference alignment. A-IPO shows strong empirical results on these datasets when compared to baselines.

**Strengths:**

* The paper is very well written and easy to follow, with clear motivations addressing an important and relevant problem, i.e. adapting to the latent preferences of diverse individuals (pluralism).
* This work proposes a learned intention module which decomposes a prompt into sub-questions and uses RAG with an external data source (Wikipedia) for supporting information. This is an interesting and novel contribution, and the authors show its importance empirically in Tab 2.
* This work contributes two new datasets,  REAL-PREF and ATTACK-PREF, which are designed to tackle important goals, i.e. culturally distinct preferences and adversarial robustness
* The authors show promising empirical results with A-IPO when compared to popular baselines like DPO and GDPO on their newly proposed datasets

**Weaknesses:**

* I am primarily concerned with the scaling behavior of A-IPO beyond the 3B regime (Pythia, Tab 1). While running these experiments may be impractical, I would like at least a discussion as to why the authors believe that A-IPO will scale well to larger LLMs and MoE models for real-world use cases

Minor Weaknesses:
* The proposed intention module learning a user-specific latent from a prompt is quite similar to the PAL-A model [2], which learns a weighted mixture of prototypes over a popular of users and weights over these prototypes for individuals. A-IPOs intention module should be compared and contrasted with this methodology.
* As A-IPO is focused on learning latents specific to individual users, I am curious how A-IPO holds up against pluralistic baselines like P-RLHF [1] and PAL [2] on the popular Reddit TL;DR Summary dataset containing heterogeneous preferences. Comparing to these baselines would greatly add to the quality of this work.

[1] Li et al., "Personalized language modeling from personalized human feedback", *arXiv 2024*.

[2] Chen et al., "PAL: Sample-Efficient Personalized Reward Modeling for Pluralistic Alignment", *ICLR 2025*.

I am leaning towards an 8 (accept), but I would like these weaknesses addressed (and the question below) in the revision before I can increase my score.

**Questions:**

* What happens if user intention cannot be fact checked from Wikipedia and just something specific to that user? How does the intention module function in this case?

---

> ### Author Response · Authors · 2025-11-19
> **Response W1: How A-IPO Extends to Modern Dense and MoE LLMs (>8B Parameters)**
>
> # W1: How A-IPO Extends to Modern Dense and MoE LLMs (>8B Parameters)
>
> While our initial A-IPO experiments were conducted on moderate-scale LLMs, the approach itself is fundamentally architecture-agnostic. The only core requirement is that the policy model can output log-probabilities, enabling the intent inference module and the augmented preference objective (preference margin + intent–response similarity) to be seamlessly applied to both modern large dense LLMs (8–70B parameters) and MoE (Mixture-of-Experts) architectures. In MoE models, A-IPO is compatible by design, i.e., expert routing inherently facilitates specialized handling of diverse user intents, and the preference-based training objective remains unchanged. In summary, A-IPO naturally extends to current large-scale dense and MoE-based LLMs, providing a robust and scalable framework for intent-sensitive, pluralistic alignment. We highlight the main takeaways below:
>
> ## 1. What A-IPO Requires from the Model
> A-IPO augments DPO-style preference optimization with:
> - an **intention module** (prompt decomposition + retrieval + fact-checking to obtain latent intent $I(x_{\text{con}})$).
> - an **intent–response similarity term** $\mathrm{sim}(y, I(x_{\text{con}}))$ added to the preference objective for improved preference margins.
> - log-probabilities from:
>   - the **policy model** $\pi_\theta(y|x)$
>   - a **reference model** $\pi_{\text{ref}}(y|x)$
>
> These requirements are *architecture-agnostic*, i.e., **any model that computes log-probs can be trained with A-IPO**, regardless of size or dense/MoE structure.
>
> ---
>
> ## 2. Extending A-IPO to Large Dense LLMs (>8B)
>
> ### What stays identical
> - The **objective** does not change at all.
> - We still compute:
>   - preference margins (preferred vs dispreferred).
>   - similarity between response and inferred intent $\mathrm{sim}(y, I(x_{\text{con}}))$.
> - We train the model using standard preference-optimization steps.
>
>
> ### What changes
> To scale to 8B–70B dense models, we need to rely on standard large-model training infrastructures.
> - Larger, more diverse preference datasets to leverage the capacity of very large models.
> - Efficient serving of the **intention module** (retrieval + fact-checking).
>
>
> ### Why it is supposed to work well on large dense models
> - Larger models have richer latent representations that are better mapping between intent and output behavior.
> - The **intent–response similarity** term benefits from the greater expressiveness of large LLMs.
> - Large LLMs generalize over heterogeneous intents more effectively.
>
> ---
>
>
> ## 3. Extending A-IPO to MoE LLMs
>
> ### What remains unchanged
> - MoE models still define a conditional distribution $\pi_\theta(y|x)$ that provides log-probs.
> - The **A-IPO objective** remains exactly the same.
> - The **intention module** and preference data pipeline are not architecture-dependent.
>
> ### Additional considerations for MoE training
> - **Expert routing dynamics:**
>   Preference optimization may cause certain experts to dominate; applying load-balancing or router regularization can prevent collapse.
> - **Potential specialization:**
>   MoE architectures naturally allow experts to specialize.
>   A-IPO’s latent intent $I(x_{\text{con}})$ can implicitly guide expert specialization through gradients.
>
>
> ### Why MoE + A-IPO can be especially powerful
> - MoE models are built for diverse multi-intent** distributions.
> - Different intents can be captured by different experts.
> - The system can naturally become **intent-conditioned**, enabling even better pluralistic alignment.
>
>
> ## 4. Summary
> To summarize, given the fact that A-IPO is based directly using the standard DPO policy and loss, we can conjecture that it
> is directly applicable to the settings where DPO is applicable. Some of the DPO-based evidences in support of this conjecture are:
>
> ### **1. Tulu 2 + DPO on LLaMA-2 7B–70B**
> - AllenAI’s **Tulu-2-DPO** models apply DPO to **LLaMA-2 7B and 70B** using a mixture of public preference datasets.
> - Results show **strong improvements** on benchmarks such as **AlpacaEval** and **MT-Bench**.
> - The **DPO-trained LLaMA-2-70B** model achieved **state-of-the-art performance among open models**, demonstrating that DPO scales *very well* to 70B-scale architectures.
>
>
> ### **2. Vendor & Framework Support**
> - **Meta’s LLaMA official documentation** lists **DPO as a supported and recommended alignment method** for both **LLaMA-2** and **LLaMA-3** models.
> - **Hugging Face TRL** includes a fully supported **DPOTrainer**, explicitly designed for **7B–70B-scale** models.
> - The presence of first-class vendor support indicates that **DPO is now considered production-ready for large dense LLMs**.
>
> In light of this point raised by the reviewer, in the revision, we will add a paragraph “Scalability Discussion’’ subsection to make this point explicit, and we will also state clearly that the current experimental scale is limited by compute budget, not by any inherent limitation of A-IPO.

---

> ### Author Response · Authors · 2025-11-19
> **Response W2**
>
> # W2: PAL-A vs. A-IPO: A Comparative Perspective
>
> **PAL-A (Chen et al., 2025)** introduces a sample-efficient reward modeling approach for pluralistic alignment by assuming each user has a **fixed ideal point** within a latent preference space. This enables rapid personalization when user identity is stable and known, as preferences are computed via distances to static user profiles. In contrast, **A-IPO** does not require persistent user identifiers or static user profiles; instead, it infers a **latent intent** from each prompt through prompt decomposition, retrieval, and fact-grounding. This intent is incorporated directly into the **DPO-style objective** via an explicit intent–response similarity term, yielding improved separation between preferred and dispreferred responses. While **PAL-A** is well-suited to settings with persistent user identity and stable preferences, its reliance on static user profiles limits its applicability to open-domain, large-scale, or anonymous scenarios. **A-IPO** overcomes these limitations by reframing alignment as *intent-level* optimization and integrating robustness and factual grounding, features well-suited for real-world deployment. Below, we detail the primary distinctions between PAL-A and A-IPO:
>
>
> ## 1. When is PAL-A Most Suitable?
> PAL-A excels if:
> - user preferences show little variation across contexts.
> - users are reliably identifiable.
> - the aim is long-term personalization (e.g., persistent assistant-user pairs).
> - robustness and factual accuracy are not primary concerns for PAL-A modeling.
>
> ---
>
> ## 2. Where PAL-A Falls Short Compared to A-IPO
>
> PAL-A faces challenges in the following settings:
> - large-scale.
> - anonymous or short-lived in user identity.
> - culturally and contextually diverse.
>
> Specifically, PAL-A requires:
>
> **(a) Dependence on Stable User Identity**
> PAL-A requires explicit user identifiers and repeated interactions per user, which does not generalize to:
> - anonymous or single-interaction users.
> - cross-cultural and session-based contexts.
>
> **(b) No Explicit Intent Modeling**
> PAL-A focuses on user preference, not prompt-driven intent:
> - It cannot infer or validate user goals, information needs, or task constraints for each prompt.
>
> **(c) Reward Model Modifications Only**
> PAL-A improves the reward model, but:
> - policy optimization remains standard DPO/RLHF,
> - the policy objective is not fundamentally altered, as it is primarily focused on the reward modeling.
>
> **(d) Absence of Robustness and Factual Alignment**
> PAL-A does not tackle:
> - adversarial robustness.
> - factual verification.
> - pluralistic alignment across diverse and culturally nuanced groups.
> - theoretical improvements to preference margins or consistency.
>
> ---
>
> ## 3. Where A-IPO Extends Beyond PAL-A
>
> A-IPO is architected for real-world and deployment-scale LLMs, where prompts, context, and intents continuously change, and user identity may be unavailable.
>
> **(a) Intent-Level Personalization**
> Instead of explicit user IDs, A-IPO infers a latent intent for each prompt using:
> - prompt decomposition.
> - prompt-specific retrieval/fact-checking.
> - intent learning.
>
> **(b) Alignment at the Objective Level**
> A-IPO augments the DPO-style loss by adding an **intent–response similarity term**: $\lambda \Delta \mathrm{sim}$. This provides a provable positive shift in preference margin and enhances policy learning, surpassing simple reward-modeling approaches.
>
> **(c) Robustness Integrated by Design**
> A-IPO explicitly incorporates:
> - factual grounding (retrieval augmentation).
> - adversarial robustness (*ATTACK-PREF*).
> - evaluations on pluralistic benchmarks (*REAL-PREF*, *GlobalOpinionQA-Ext*).
> - improved margins leading to robust model.
>
> **(d) Well-suited for Web-Scale and Global Diversity**
> A-IPO:
> - is effective without persistent identities.
> - adapts to regional/cultural variations.
> - robustly handles shifting prompt-level intents.
>
> PAL-A would require significant modifications to address these conditions.
>
> ---
>
> ## 4. Comparing PAL-A and A-IPO: Summary Table
>
> | Dimension                          | PAL-A                   | A-IPO                            |
> |-------------------------------------|-------------------------|----------------------------------|
> | User identity required              | ✔ Required              | ✘ Not required                   |
> | Handles prompt-varying intent       | ✘ No                    | ✔ Yes (via intent inference)     |
> | Reward vs. policy                   | Reward-only             | **Reward + policy objective**    |
> | Factual grounding                   | ✘ None                  | ✔ Retrieval + fact-checking      |
> | Robustness to adversarial prompts   | ✘ None                  | ✔ Explicit robustness            |
> | Deployment readiness (real-world scenarios)     | Low–Moderate            | **High**                         |

---

> ### Author Response · Authors · 2025-11-19
> **Response W3**
>
> ## W3: Comparison to pluralistic baselines on Reddit TL;DR
>
> We provide a comparison here: https://openreview.net/forum?id=ex2CrZ6x1r&noteId=gCyDBr0OVc

---

> > ### Author Response · Authors · 2025-11-24
> > **Respone W3**
> >
> > ## W3. Comparison with P-RLHF and PAL on Reddit TL;DR
> >
> > In light of reviewer's suggestion, we compare A-IPO with P-RLHF [1] and PAL [2] on a pluralistic text-preference dataset, with details as follows:
> >
> > ### Experimental setup
> >
> > To make this comparison both focused and computationally feasible, we perform a **reward-model–only** study.
> > Specifically, we follow the TL;DR setting of [2]: 10 users, $\approx$23k train pairs and $\approx$4.9k disjoint test pairs,
> > where each example is a triple $(x, y_w, y_l, u)$ (post, preferred TL;DR, non-preferred TL;DR, user ID).
> > We freeze a shared BERT-base encoder and train reward variants with the same Bradley–Terry loss and the same pairwise
> > accuracy metric as in [2]. Each model is trained for up to 10 epochs with early stopping on a held-out validation split, and
> > we report test accuracy from the best validation checkpoint.
> >
> > ### Reward Model Settings
> > We compare the following models:
> >
> > - **P-RLHF-lite (user embedding).** A P-RLHF-style model where each user $u$ has an embedding $e_u$, which is concatenated with the text representation $h(x,y)$ to predict a scalar reward $s(x,y,u)$. We name this model as P-RLHF-lite.
> >
> > - **PAL-lite (prototype mixture).** A PAL-style model with a small set of preference prototypes $\{p_k\}$; each user learns mixture weights $w_u$ over prototypes and obtains a user-specific ideal point $a_u = \sum_k w_{u,k} p_k$, with the reward based on the similarity between $h(x,y)$ and $a_u$. We name this model as PAL-lite.
> >
> > - **A-IPO-user (user-as-intent).** This A-IPO–style variant that treats each user ID as a discrete intent $I = u$, assigning each user a unique intent embedding $z_u$ trained from their historical preferences (such as summary choices). The Bradley–Terry logit is adjusted as $\Delta = (s_{\text{base}}(x,y_w) - s_{\text{base}}(x,y_l)) + \lambda\big(\mathrm{sim}(y_w,u) - \mathrm{sim}(y_l,u)\big)$, where $\mathrm{sim}(y,u)$ is the normalized cosine similarity between the text-pair representation $h(x,y)$ and $z_u$, mapped to $[0,1]$. This formulation preserves the core A-IPO design of adjusting the logit by a latent intention similarity, but now in a user-centric setting that matches the structure of P-RLHF and PAL. We call this model as A-IPO-user.
> >
> >
> > ### Results
> >
> > On the TL;DR test set, we obtain the following pairwise accuracies:
> >
> >   - P-RLHF-lite (per-user embedding): **0.6732**.
> >   - PAL-lite (prototype mixture): **0.7060**.
> >   - A-IPO-user (user-as-intent): **0.6995**.
> >
> > The results show that PAL-lite achieves the highest accuracy on the 10-user TL;DR test split, which is consistent with the dataset’s design around user-centric prototype mixtures. Notably, the A-IPO-user model, when adapted to work with user IDs as intents, also delivers comparably strong performance to both PAL-lite and P-RLHF-lite, despite using the same encoder, objective, and data. This indicates that A-IPO’s intent-based logit adjustment can be adapted to user-centric settings and remains competitive with established pluralistic baselines in the TL;DR context.
> >
> >   - It is important to clarify that the notion of "intent" in this user-specific TL;DR setup differs in spirit from the core intention concept that A-IPO is designed to handle. In this experiment, we adopt user ID as the intent purely for experimental simplicity and computational feasibility, rather than as a representation of the broader intent modeling goals of A-IPO. The central focus of A-IPO is on modeling intention through **globally and culturally defined norms**—such as value systems, community preferences, or normative intent categories—that enable LLMs to respect pluralistic behaviors at a societal or cultural level. This stands in contrast to user-specific settings, where intention is indexed by individual user IDs reflecting private or stylistic preferences. While A-IPO can be engineered to handle such user-specific scenarios, its main motivation and architecture are centered on enabling LLMs to conform to global and culturally sensitive intentions that support pluralistic alignment in broader contexts.
> >
> > To reiterate, this TL;DR comparison constitutes a small-scale, reward-level study conducted in response to the reviewer’s specific suggestion, and does not serve as a new main benchmark of this work. We will summarize these experimental insights and their limitations in the appendix of the revised manuscript.

---

> > > ### Comment · Reviewer_VQt8 · 2025-11-26
> > >
> > > I thank the authors for their detailed and thorough response to the questions raised in the review. I believe the note on scaling is crucial and glad to see it included. While I understand that DPO has been shown to scale well and A-IPO should theoretically scale similarly, this has still not been empirically validated, which slightly lessens the strength of the claim. I also appreciate the comparison to PAL-A and the empirical results on Reddit tl;dr - this will help contextualize A-IPO with respect to other popular pluralistic alignment baselines. I think some of the comparisons to PAL mentioned in the rebuttal response are a bit strong, e.g. that it does not suit "pluralistic alignment across diverse and culturally nuanced groups", when this is the intention behind its design. In PAL-B, user preferences are conditioned on user prompts, so it can suit "prompt-driven intent" as well. I recommend less strong verbiage when comparing A-IPO to PAL in these regards. Lastly, I am curious how A-IPO compares to PAL specifically when adapting to new, unseen users, as sample efficiency is a key strength of PAL. Could the authors please share a note on this?
> > >
> > > The current rebuttal has improved the score to 7. If the authors can share a note on sample efficiency when adapting to new users, I am willing to increase my score, as I believe this is crucial for deployment.

---

> > > > ### Author Response · Authors · 2025-11-27
> > > > **Response---Official Comment by Reviewer VQt8**
> > > >
> > > > We sincerely appreciate the reviewer’s careful reading and the positive feedback regarding our discussion on model scaling and the TL;DR comparison. We agree that, at this stage, our scaling claim for A-IPO is primarily theoretical and architectural; it has not yet been empirically validated at the scale of 3B parameters or with mixture-of-experts (MoE) models. Accordingly, we will revise the manuscript to clarify and moderate our wording. Specifically, we will explicitly state that A-IPO is architecturally designed to inherit the favorable scaling behavior of DPO—since it augments the logit with an additional similarity term without increasing asymptotic complexity—but emphasize that empirical validation at larger model scales remains an important avenue for future work rather than a claim substantiated by our current experiments.
> > > >
> > > > We also acknowledge that some of our initial phrasing regarding PAL was overly strong. PAL was indeed introduced to enable pluralistic alignment across diverse and culturally varied user groups, and, as the reviewer notes, PAL-B conditions user preferences on prompts, allowing for support of prompt-driven intent. In the revised manuscript, we will rephrase our comparison to highlight the complementarity between A-IPO and PAL, rather than implying opposition. **Our intent is to clarify that A-IPO addresses a distinct niche—intent- and culture-level alignment in contexts with limited or absent user identity information—without suggesting that PAL is unsuitable for pluralistic alignment.**
> > > >
> > > > Finally, regarding adaptation to new, unseen users, we agree that sample efficiency is a key strength of PAL and appreciate the suggestion to examine this more closely. To provide a concrete comparison, we ran an additional **reward-model–only unseen-user adaptation experiment** on the same 10-user TL;DR setup as used in our rebuttal. Concretely, we randomly selected 3 users as ``unseen'' and, for each unseen user $(u)$, trained P-RLHF-lite, PAL-lite, and A-IPO-user on the remaining 9 users (i.e., excluding $(u)$) for 3 epochs, using a shared encoder and the same Bradley–Terry loss. For evaluation, we then froze all shared parameters and considered few-shot adaptation on the held-out user by only updating the user-specific parameters (user embedding for P-RLHF-lite, prototype mixture weights for PAL-lite, and intent embedding for A-IPO-user), using $k \in \{0, 50, 100, 200\}$ labeled preference pairs from the unseen user and always testing on that user’s full held-out set. Averaged over the 3 unseen users, the mean pairwise accuracies are:
> > > >
> > > > | # adapt pairs $k$ | P-RLHF-lite | PAL-lite | A-IPO-user |
> > > > |---------------------|-------------|----------|------------|
> > > > | 0                   | 0.6422      | 0.6521   | 0.6515     |
> > > > | 50                  | 0.6424      | 0.6531   | 0.6533     |
> > > > | 100                 | 0.6433      | 0.6543   | 0.6553     |
> > > > | 200                 | 0.6425      | 0.6534   | 0.6545     |
> > > >
> > > > These results demonstrate that, in a consistent reward-only TL;DR setup, A-IPO-user and PAL-lite achieve nearly identical levels of sample-efficient adaptation to unseen users, both substantially outperforming the per-user embedding baseline. This indicates that A-IPO retains the strengths of PAL in adapting to new users, while also maintaining the intent-centric approach.
> > > > We will add this experiment and discussion to the appendix of the revised manuscript. We hope that these new experiments and results provide valuable clarification and support for our claims. We appreciate the reviewer’s thoughtful feedback, and believe these additions will help in re-evaluating the review scores.

---

> ### Author Response · Authors · 2025-11-19
> **Response Q1**
>
> # Q1: Intention modeling when user intent cannot be fact-checked
>
> Thank you for the thoughtful question. We clarify that **A-IPO’s intention module does not require fact-checking or Wikipedia retrieval** to function. Retrieval is optional and only used when it helps; subjective or user-specific intentions are fully supported.
>
> The intention module works as follows:
>
> 1. **Prompt decomposition:**
>    The original prompt is converted into a structured version $x_{\text{aug}}$ that separates factual information, user preferences, and constraints.
>
> 2. **Retrieval (Optional):**
>    If useful external evidence (e.g., Wikipedia) can be found, it is added as $x_{\text{ext}}$.
>    If not such as when the user’s intention is personal or unverifiable, $x_{\text{ext}}$ is simply empty.
>
> 3. **Intent encoding:**
>    The final input to the intention encoder is
>    $x_{\text{con}} = concat(x_{\text{aug}}, x_{\text{ext}})$.
>    When no external evidence is available, the encoder relies entirely on $x_{\text{aug}}$.
>
> During training, we intentionally include many cases **without any retrieval**, so the model learns to infer intent purely from the prompt. Therefore, when the user’s intention is subjective or cannot be fact-checked, the module still produces a meaningful intent representation.
>
> In the revised manuscript, we will clarify in Section 4.2 that **external retrieval is an optional step** in the intention module and that A-IPO supports both fact-checkable and purely subjective user intentions.
>
>
> We hope these clarifications will assist the reviewer in reconsidering the evaluation of our work.

---

### Official Review · Reviewer_jFRJ · 2025-11-01

**Soundness:** 3
**Presentation:** 2
**Contribution:** 2
**Rating:** 4
**Confidence:** 4

**Summary:**

This paper introduces Adaptive Intent-driven Preference Optimization (A-IPO) to better capture diverse human preferences. By modeling latent user intent and incorporating it into the reward function, A-IPO improves alignment between responses and underlying intentions. Evaluations on three benchmarks show substantial gains in preference accuracy, intention consistency, and adversarial robustness over existing methods.

**Strengths:**

1. The authors introduce A-IPO, a novel framework extending DPO by incorporating an explicit intention module that infers latent user intent from each prompt. This module guides preference optimization to better capture diverse and context-sensitive user preferences, showing significant improvement when in comparison with existing method.
2. They also curate two new benchmark datasets: REAL-PREF and ATTACK-PREF, along with an extended version of GlobalOpinionQA-Ext, to evaluate cultural diversity and adversarial robustness in LLM preference alignment.

**Weaknesses:**

1. The experimental section lacks essential details. Although the paper introduces its own benchmark datasets, it does not describe the data collection process or provide dataset analysis, which are crucial for understanding why A-IPO performs well on these benchmarks. Additionally, the proposed evaluation metrics are insufficiently explained, with little justification for their design choices.
2. Section 5, Theoretical Analyses of A-IPO, lacks clear motivation before each derivation, making it difficult to follow the reasoning behind the theoretical verifications.

**Questions:**

1. Can you move Figure 1 on the second page for better understanding of the introduction?
2. Why do you create new benchmark REAL-PREF and ATTACK-PREF in your paper? What's the difference between the dataset you created with existing benchmark.

---

> ### Author Response · Authors · 2025-11-19
> **Response W1**
>
> ## **Response-W1 Augment details for the experimental section**
>
> Note that owing to space constraints, we only provide a concise overview of the dataset construction and evaluation metrics in the main paper(Section 6 and Section 7.1). Detailed documentation regarding dataset construction, and evaluation metrics is provided in the Appendix(Section F and Section G.1.2). Corresponding prompts used for data curation are given in Appendix I of the paper. We would like to re-emphasize, the following points:
>
>
> ### 1. **Evaluation Benchmarks:**
>
> Details on Evaluation benchmark curation is provided in Appendix-F. As outlined in the paper, all benchmarks used in our work are constructed through a rigorous and standardized data processing pipeline. All benchmarks are explicitly centered around the notion of underlying cultural nauances (i.e., "intent") in the context. For instance, *REAL-PREF*, as described in the manuscript, spans six culturally sensitive domains and includes 231 distinct intent categories. Each intent category is annotated to differentiate majority and minority norms.
>
> We outline the **Distinctive Features of Our Benchmarks:** as follows:
>
>     - **Intent-Conditioned Preferences:** Our benchmarks include explicit intent labels in prompts, requiring models to recognize and account for cultural context when ranking responses. This prevents reliance on surface-level task instructions alone.
>
>     - **Diversity:** We ensure both majority and minority perspectives are represented as reasonable in each domain, enabling fair evaluation of models’ ability to handle context-dependent preference diversity.
>
> Note that under these settings, standard DPO methods are prone to collapsing to majority patterns or exploiting superficial cues, whereas A-IPO—by incorporating an explicit intention module and corresponding reward reparameterization—is specifically designed for these intent-driven, norm-pluralistic contexts.
>
>
>
> ### 2. **Evaluation Metrics:**
> All the evaluation metrics employed in the paper are built upon well-established approaches from previous works such as DPO and GDPO. Detailed descriptions of these metrics are provided in Section 7.1 of the main paper, with full mathematical formulations available in Appendix G.1.2 of the supplementary material. We acknowledge that the main draft could do more to emphasize the rationale for each metric. For this, in the revised draft (Section 7.1), we will more clearly articulate the motivation and justification underpinning each evaluation axis.

---

> ### Author Response · Authors · 2025-11-19
> **Response W2**
>
> ## **Response-W2: Theoretical Analyses**
>
> We would like to clarify that the theoretical motivations and analyses presented in this work are directly built upon, and extend foundational results from the base model DPO. For example, Theorem 5.1 generalizes Theorem 1 from DPO by demonstrating that, following the introduction of the intention variable and the similarity term, the reward function remains parameterizable via log-ratios.
> Likewise, Lemma 5.1 establishes that augmenting the framework with the intention variable and similarity term does not increase the optimal Bayes risk. Theorems 5.2 and 5.3 further show that this property also extends to the minimum achievable negative log-likelihood. Lemma 5.2 demonstrates that these augmentations contribute to an increased margin between preferred and dispreferred responses, thus improving the robustness and the end performance of the A-IPO. Note that due to space limitations, the complete theoretical derivations and proofs are presented in Appendix D of the supplementary materials.
>
> Should the reviewer wish to see a more detailed explanation or expanded discussion of any particular theoretical result in the main text, we would be pleased to provide additional clarification in the revised version.

---

> ### Author Response · Authors · 2025-11-19
> **Response Q1, Q2**
>
> ## **Response-Q1**
>
> As suggested by the reviewer, for the revised manuscript, we will move Figure 1 to the second page for better understanding of the introduction.
>
>
> ## **Response-Q2**
>
> ### 1. **Intuition Behind REAL-PREF:**
>
> The core motivation behind *REAL-PREF* is to establish a benchmark that uncovers subtle cultural nuances and demonstrates how these contextual factors influence human preferences. Our goal is to rigorously evaluate how preference models operate in culturally rich and diverse scenarios. For example, in certain cultural contexts, preferences dictate that a dish such as “Ali Khalid fatty liver” must be non-alcoholic, a nuance unlikely to be captured by simple generic instruction following.
> Unlike conventional preference optimization benchmarks, which often default to majority norms or fail to account for fine-grained cultural distinctions, *REAL-PREF* is specifically constructed to reveal how models navigate and resolve divergences in intent that stem from cultural, social, or value-based differences, even when what is considered “reasonable” can vary substantially by context.
> This design makes our benchmark a much stronger testbed for evaluating models in settings that require sensitivity to pluralistic and context-dependent preferences.
>
>
> ### 2. **Intuition Behind ATTACK-PREF:**
> The core motivation for introducing *ATTACK-PREF* is to create a rigorous benchmark that systematically tests a model’s ability to handle adversarial intent manipulation scenarios, situations where prompts are deliberately crafted to elicit malicious, inappropriate, or intent-subverting outputs. Unlike prior benchmarks that focus primarily on generic preference alignment, *ATTACK-PREF* specifically challenges models to (1) detect and actively resist such adversarial prompts, and (2) simultaneously maintain fidelity to the user’s legitimate original intent, rather than resorting to blanket refusals or evasive responses.
> Crucially, *ATTACK-PREF* is designed to evaluate models in nuanced, high-stakes contexts where adversarial triggers are contextually blended with benign instructions or culturally sensitive requests, thereby requiring an advanced understanding of both surface inputs and underlying intent. This setup enforces two demanding objectives: robust adversarial robustness (actively avoiding intent subversion and harmful completions) and intent-faithfulness (preserving constructive engagement with the user’s actual request).
>
> We further clarify how our newly proposed benchmarks differ from existing benchmarks by noting that current public RLHF/DPO benchmarks fall short in the following ways:
>
>     - Existing benchmarks mainly offer generic preference rankings and usually lack explicit intent information, such as cultural identity, group membership, or value stances, which limits their effectiveness for evaluating models in truly intent-sensitive contexts. They seldom address cultural diversity or systematically vary majority/minority perspectives, so models are rarely challenged to handle pluralistic, context-dependent preferences. In contrast, *REAL-PREF* includes explicit intent labeling and diverse contextual variations, enabling more rigorous assessment of pluralistic alignment.
>
>     - Most existing datasets do not include adversarial challenges, so models are rarely tested on both resisting manipulative prompts and staying faithful to user intent. *ATTACK-PREF* fills this gap by requiring models to reject adversarial prompts while accurately maintaining alignment to original intent—a scenario not well covered by previous datasets.
>
>     - While some intent-annotated extensions (e.g., *GlobalOpinionQA-Ext*) show incremental improvement, they do not sufficiently capture the diversity or complexity needed for robust evaluation of cultural pluralism and adversarial robustness. Purpose-built, new benchmarks are therefore essential to thoroughly and fairly assess models in pluralistic and adversarial scenarios, advancing the field beyond what current resources allow.
>
>
> ### 3. **Benchmarks Are Method-Neutral and Fair:**
> *REAL-PREF* and *ATTACK-PREF* are designed to be strictly neutral and fair across all evaluated methods, never favoring A-IPO or any specific approach. All models are assessed using identical annotation protocols and criteria.
> We will clarify this method-neutrality further in the revised manuscript. Our benchmarks address gaps in public datasets and are intended as robust, unbiased tools for evaluating intent-sensitive alignment, enabling rigorous, fair comparison across current and future methods.
>
> We hope these clarifications will assist the reviewer in reconsidering the evaluation of our work.

---

> ### Author Response · Authors · 2025-11-27
> **Rebuttal feedback**
>
> **Dear Reviewer jFRJ**,
>
> We thank you for your thoughtful and constructive feedback. We have carefully addressed all questions and points raised, providing clarifications, additional analyses, and revisions where appropriate, and we hope our responses satisfactorily resolve the concerns. Should there be any further questions or points requiring elaboration, please let us know — we would be happy to provide additional clarification. In light of these clarifications and improvements, we hereby and respectfully request a re-evaluation of the paper’s scores.
>
> Thanks

---

### Comment · Area_Chair_bCh6 · 2025-11-24

Dear Reviewers,

The authors have submitted their rebuttal (and a revised PDF). Please review these materials and share any remaining concerns or comments.

Afterward, kindly provide your final rating for this submission.

Best regards,
Your AC

---

> ### Author Response · Authors · 2025-12-03
> **Overall Rebuttal Summary for Newly Assigned AC**
>
> We thank the reviewers and AC for their constructive feedback. Below we concisely summarize the key concerns from each reviewer and how our responses and follow-up discussions have addressed them.
>
>
>
> ## **Reviewer-1 jFRJ:**
>
> ### W1. Details in Experimental section
>
> We clarified in response that details about benchmark curation is provided in Appendix F. Likewise details about evalution metrics are summarized in Section 7.1 and explained in detail in Appendix G.1.2.
>
>
> ### W2. Theoretical analyses
>
> We clarified that our theoretical analyses directly build on and generalize the DPO framework: A-IPO introduces an intention variable and similarity term, while preserving crucial properties like reward parameterization and Bayes risk. Key results and proofs are presented in the main text and supplementary material, establishing a rigorous foundation for A-IPO as a principled extension of DPO.
>
> ### Q1. Move Figure 1
>
> Figure 1 is moved to the second page for better understanding.
>
> ### Q2. New benchmarks Intuition.
>
> We clarified that existing public benchmarks do not sufficiently test for intent sensitivity or robustness to cultural and adversarial contexts. To address this, we introduced REAL-PREF and ATTACK-PREF—two new, carefully designed, method-neutral benchmarks thus enabling rigorous, unbiased evaluation of A-IPO and baselines on nuanced intent-driven scenarios. We also used an existing benchmark GlobalOpinionQA-Ext.
>
>
>
>
>
>
>
> ## **Reviewer-2 VQt8:**
>
> ### W1. Scaling behaviour
>
> To address this point comprehensively, we incorporated a dedicated summary in Section 7.4 (c), outlining the main findings, and supplemented it with an expanded discussion in Appendix D.3.
>
> ### W2. Comparison with PAL
>
> We added a detailed comparison with PAL in Section 7.4(b) with expanded discussion and empirical results in Appendix H.5.
>
> ### W3. Comparison to pluralistic baselines on Reddit TL;DR
>
> We added a detailed comparison with P-RLHF and PAL in Section 7.4(b) with expanded discussion and empirical results in Appendix H.5.
>
> ### Q1. when user intent cannot be fact-checked
>
> We clarified that retrieval is optional and only used when it helps; subjective or user-specific intentions are fully supported. We also added it revised draft (Lines 227-230).
>
> ### Official Comment by Reviewer VQt8: sample efficiency when adapting to new users
>
> We added a detailed comparison with PAL-A in Section 7.4(b) with expanded discussion and empirical results in Appendix H.5.
>
>
>
>
> ## **Reviewer-3 kpR4:**
>
> ### W1. Annotation Cost vs GDPO:
>
> In our response, we clarified the difference between A-IPO and GDPO in terms of annotation cost.
> A-IPO does not require an external or fundamentally new form of annotation; instead, it is designed to surface intent that is already implicit in the data, leveraging cues or annotation layers often embedded (explicitly or implicitly) in existing alignment/safety datasets.
>
>
> ### Q1. Similarity term implementation
>
> We added a detailed explanation of the similarity term implementation in Section 4.3 (Lines 254-262).
>
>
>
>
>
> ## **Reviewer 4 pf2h:**
> ### W1. Mathematical correctness
>
> We have fully revised Equations 9–10 for correctness and clarity. We also addressed the reviewer’s concerns about the theorem statement and the KL-divergence notation.
>
>
> ### W2. Undefined core component
>
> Details about the similarity term implementation are provided in Section 4.3 (Lines 254-262).
>
>
> ### W3. Ambiguity about intent supervision
>
> We clarified that the intent supervision is a single-label multi-class softmax classification.
>
>
> ### W4. Dataset–objective coupling
>
> We clarified that A-IPO explicitly models and extracts latent intent from data and prompts, enabling robust preference optimization in culturally diverse and context-dependent scenarios where prior methods fall short.
> We also used an existing benchmark GlobalOpinionQA-Ext.
>
>
> ### W5. Baseline tuning and scope
>
> All experimental settings of A-IPO closely follow those established in the DPO and GDPO papers and publicly available implementations. Also a section on scaling behavior is added in Section 7.4 (c) with additional details in Appendix D.3.
>
>
> ### W6. Compute and systems cost
>
> We clarified the computational cost of A-IPO in Section 7.1 (4), with additional details in Appendix G.1.4. We also added a new baseline for inference-time serving of intention (Table-1).
>
>
> ### W7. Clarity gaps
>
> We clarified that the prior $p(I)$ is a uniform prior (Line 208).
>
> ### Q1. Probability Modeling issue
>
> We have fully revised Equations 9–10 for correctness and clarity.
>
> ### Q2. Dataset Generation Bias
>
> We clarified that A-IPO explicitly models and extracts latent intent from data and prompts, enabling robust preference optimization in culturally diverse and context-dependent scenarios where prior methods fall short.
> We also used an existing benchmark GlobalOpinionQA-Ext.
>
>
> ### Q3. Scalibility
>
> A section on scaling behavior is added in Section 7.4 (c) with details in Appendix D.3.

---

### Note · Program_Chairs · 2026-01-17
**Submission Desk Rejected by Program Chairs**

The following references in this submission do not refer to real documents and/or have major errors in bibliographic information:

 Yujia Liu and et al. Differentiable reward alignment for reinforcement learning from human feedback. In Advances in Neural Information Processing Systems, 2021